# COMPACT TEXT-TO-SDF VIA LATENT MODELING

## ABSTRACT

This paper introduces CDiffSDF, a lightweight Text-to-Shape model designed for efficient 3D shape generation. By harnessing latent-code-based signed distance functions (SDFs), CDiffSDF not only produces high-resolution shapes but also features diffusion denoising capabilities within the latent space. Its generation ability is further boosted by integrating Gaussian noise during the SDF training phase, effectively counterbalancing the diffusion sampling perturbations. Transitioning from the core concept of Text-to-SDF, our model is versatile, as it can seamlessly adapt and generate shapes influenced by a range of inputs, including text, class, and image conditions. Experimental results demonstrate CDiffSDF's ability to produce detailed shapes, all within a compact design.

## 1 INTRODUCTION

The ability to translate textual descriptions into corresponding 3D structures can revolutionize various sectors, from aiding design professionals to enhancing learning experiences in education. Imagine the convenience for a designer if a mere text prompt could yield intricate 3D structures or the enhancement in education if kids could visualize complex geometries simply from their descriptions (Fig. 1 Left). However, for widespread adoption, especially on devices with limited computational resources, there is a pressing need for models that are compact. Yet, this compactness should not compromise their capability to generate detailed geometry—a balance that is challenging but pivotal.

Current Text-to-Shape models generally fall into two categories. The first category consists of models that adapt 2D image-text techniques like CLIP (Radford et al., 2021) and Imagen (Saharia et al., 2022). These models can manage basic 3D object generation (Jain et al., 2021; Poole et al., 2022; Lin et al., 2023) but grapple with intricate geometric representations, especially when given detailed text prompts (Luo et al., 2022; Qian et al., 2023). The second category includes models that train directly on 3D-text pairs. Within this, some methods employ direct 3D representations such as Point Clouds (Zhou et al., 2021; Nichol et al., 2022) or Voxels (Chen et al., 2018; Liu et al., 2022), leading to a compromise in the resolution and detail of the generated shapes. Meanwhile, other models in this category opt for heavy backbones (Li et al., 2023; Jun & Nichol, 2023; Cheng et al., 2023) (more than 400M parameters). These hefty models, despite their potential accuracy, are computationally demanding and less suited for lightweight applications.

To overcome the above limitations, we utilize latent-code based signed distance functions (SDFs) (Park et al., 2019). This choice elevates the resolution and detail of generated shapes, overcoming the limitations of using fixed-resolution 3D representations. Additionally, the compactness of the latent SDF representation (256-dims versus 1,048,576-dims in Shap-E (Jun & Nichol, 2023)), allows efficient application of the diffusion objective in the latent space. A key feature of *CDiffSDF* is its robustness to noise, achieved by adding Gaussian noise during the training phase of our SDF generative model. This makes the model resilient to perturbations that can occur during diffusion sampling. Additionally, we have enhanced SDF modeling performance with reduced hyperparameters.

To evaluate geometry generation capabilities, we tested on 3D-text datasets with detailed structural descriptions and devoid of color information (Text2Shape, ShapeGlot, and ABO (Chen et al., 2018; Achlioptas et al., 2019; Collins et al., 2021)). Results show that *CDiffSDF* performs competitively in fidelity, quality, and diversity when compared to text-to-shape baselines, while maintaining a small model size. Additionally, it can be easily extended to support other input-conditional generation (Fig. 1 Right). Limitations, failure cases, and future work are discussed in Appendix A.

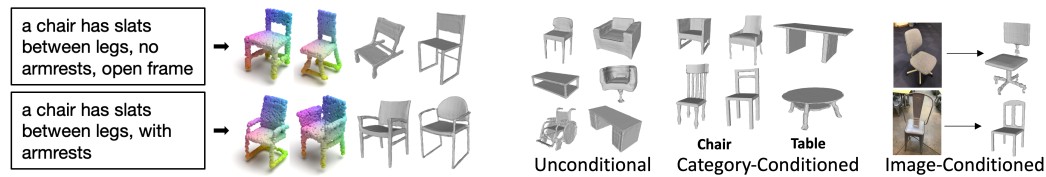

Figure 1: Shape Generation by *CDiffSDF*. **Left**: Text-to-SDF results. we render SDFs via both meshes (reconstructed from occupancy voxel grids of size $256^3$) and point clouds (2,048 points). **Right**: CDiffSDF can be enhanced for broader input-conditioned 3D shape generation.

## 2 RELATED WORK

**3D Shape Generative Modeling**: Several renowned generative models such as generative adversarial network (Goodfellow et al., 2014; Wu et al., 2016; Achlioptas et al., 2018; Zheng et al., 2022; Zhang et al., 2023b), variational autoencoder (VAE) (Hinton & Salakhutdinov, 2006; Gadelha et al., 2018), auto-regressive models (Van Den Oord et al., 2016; Luo et al., 2020; Cheng et al., 2022), and flow-based models (Rezende & Mohamed, 2015; Yang et al., 2019) have made significant strides in 3D generation tasks despite facing issues like unstable training and model collapse. The emergence of diffusion models in deep generative models (Sohl-Dickstein et al., 2015; Ho et al., 2020; Song et al., 2020) holds promise for addressing these challenges. They have achieved notable advancements (Ho et al., 2020; Ramesh et al., 2022) and have been widely used in generating point clouds (Lyu et al., 2021; Luo & Hu, 2021; Zhou et al., 2021; Zeng et al., 2022), meshes (Liu et al., 2023c; Gupta et al., 2023), and neural fields (Hui et al., 2022; Shue et al., 2023; Zhang et al., 2023a; Erkoç et al., 2023). CDiffSDF also utilizes diffusion objectives within the latent space to model the signed-distance functions (Park et al., 2019), offering a notable reduction in latent space size (256 vs. 1,048,576) compared to Jun & Nichol (2023). Furthermore, the simplicity of our framework stands out against Chou et al. (2022), which incorporates extra VAE modules.

**Text-to-Shape Synthesis**: Text-to-Shape methods predominantly align with two main categories. The first category (Jain et al., 2021; Poole et al., 2022; Lin et al., 2022; Sanghi et al., 2022; Zhu & Zhuang, 2023; Wang et al., 2023; Chen et al., 2023b; Lorraine et al., 2023) significantly depends on 2D-text pre-trained models (Radford et al., 2021; Saharia et al., 2022; Rombach et al., 2022) through the process of score distillation sampling. These methods exhibit competence in managing basic 3D objects without requiring 3D supervisions. Nonetheless, they lack the ability to efficiently handle text prompts containing geometric details (Luo et al., 2022; Qian et al., 2023), and their dependence on large-scale 2D-text pre-trained models results in substantial model sizes. In contrast, CDiffSDF does not employ any image-text pre-trained models. It focuses on training with 3D-text pairs, a notable difference from approaches like Xu et al. (2023) which adopts both 2D and 3D priors.

The second category mainly utilizes available 3D-Text pairs (Chen et al., 2018; Achlioptas et al., 2019; Collins et al., 2021; Deitke et al., 2023; Luo et al., 2023b). It directly trains on (explicit 3D representation, Text) pairs such as Point Clouds (Nichol et al., 2022; Luo et al., 2023a) and Voxels (Chen et al., 2018; Liu et al., 2022), or (implicit 3D representation, text) pairs like implicit function (Jun & Nichol, 2023) and grids of SDF (Li et al., 2023). Those utilizing explicit 3D representation face challenges with fixed and generally low resolution due to computational limitations, leading to the inability to preserve detailed structures. Conversely, methods using implicit representations show excellent performance but typically have extensive backbones and pre-trained encoders, amounting to more than 400M parameters. CDiffSDF utilizes a latent-code-based SDF, overcoming the constraints of fixed resolution and presenting enhancement beyond the $64^3$ resolution utilized in SDF grids (Li et al., 2023). For simplicity, this study employs the SDF formation from Park et al. (2019), without exploring further advanced frameworks at this stage (Genova et al., 2020; Tretschk et al., 2020; Li et al., 2022; Tang et al., 2021; Mescheder et al., 2019; Liu et al., 2021; Peng et al., 2020; Martel et al., 2021; Takikawa et al., 2021). The encoder and the latent diffusion model in CDiffSDF are both lightweight, which contributes to the compactness of the overall model. This feature sharply contrasts with the heavy VQVAE and UNet diffusion model utilized in Cheng et al. (2023).

This paper focuses on geometry study excluding texture and colors, and thus, less connected to paper studies Text-to-Texture over 3D shapes (Michel et al., 2022; Wei et al., 2023; Chen et al., 2023a). Additionally, we do not leverage image-to-3D (Nichol et al., 2022; Liu et al., 2023b;a; Melas-Kyriazi et al., 2023; Tang et al., 2023; Shi et al., 2023), but directly modeling text-to-shape.

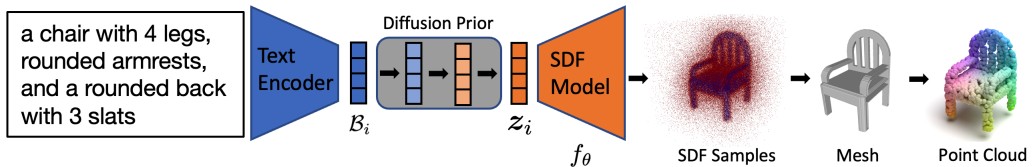

Figure 2: **Overview of CDiffSDF inference**. We begin by converting the input text prompt into embeddings, $\mathcal{B}_i$, used as conditions in our learned diffusion model to predict a latent code, $z_i$. This code is input into our SDF model $f_\theta$ to generate SDF samples, $S$, represented as coordinate pairs $c_j$ and corresponding SDF values $s_j$. The SDF values indicate whether sample points are inside ($s_j < 0$, represented in blue) or outside ($s_j > 0$, represented in red) the shape. We then reconstruct the mesh using the SDF samples through the marching cube, and sample point clouds from the mesh to compute point-cloud-based metrics in our experiments.

## 3 METHOD

This section presents *CDiffSDF*, our approach that utilizes a latent diffusion probabilistic model for Text-to-Shape. We begin by outlining the formulation of the latent-code-based signed distance function (SDF) and its connection with diffusion priors (Section 3.1). Next, we demonstrate the training and sampling algorithm of CDiffSDF conditioned on text in Section 3.2. Finally, Section 3.3 introduces several techniques to enhance the shape reconstruction performance of the latent codes, including one designed for diffusion properties and others for general improvement.

### 3.1 FORMULATION

The main concept behind the latent-code-based SDF model is to train a model that can maximize the posterior distribution over a trainable latent code given SDF samples. By learning this model, we can reconstruct the shape using the learned model and latent code. We adopt a similar approach to (Park et al., 2019) and formalize the model as follows.

Given the dataset of shapes represented as signed distance functions $\{SDF_i, i = 1, \cdots, N\}$ ($N$ is the number of shapes in the dataset), we model a shape in the dataset by sampling a set of points along with their signed distance values and re-represent the shape as a set of 4-tuples: $S_i = \left\{ (c_j, s_j), c_j = (c_j^x, c_j^y, c_j^z) \right\}_{j=0}^{\mathcal{K}_{train}}$, where $c_j$ is the 3-dim coordinates for sampled query points, $\mathcal{K}_{train}$ is how many points we sample during training, and $s_j$ is the corresponding SDF value (i.e., signed distance between the query point $c_j$ and the shape surface).

For each shape, we pair with a latent code $\{z_i, i = 1, \cdots, N\}$ initialized as $p(z_i) = \mathcal{N}(0, \sigma^2 I)$ and maximize the posterior over the latent code $z_i$ given shape 4-tuples $S_i$: $p_\theta(z_i \mid S_i) = p(z_i) \prod_{(c_j, s_j) \in S_i} p_\theta(s_j \mid z_i; c_j)$, and we reparameterize the likelihood with a neural network: $p_\theta(s_j \mid z_i; c_j) = \exp(-\mathcal{L}(f_\theta(z_i, c_j), s_j))$, where $\theta$ denote the learnable parameters of the neural network. Our optimization objective can be written as:

$$\arg\min_{\theta, \{z_i\}_{i=1}^N} \sum_{i=1}^N \left( \sum_{j=1}^{\mathcal{K}_{train}} \mathcal{L}(f_\theta(z_i, c_j), s_j) + \omega \|z_i\|_2^2 \right) \tag{1}$$

where the second term is used for regularization and $\omega$ is a hyperparameter. $\mathcal{L}(f_\theta(z_i, c_j)) = |\text{clamp}(f_\theta(x), \delta) - \text{clamp}(s, \delta)|$, where $\text{clamp}(x, \delta) := \min(\delta, \max(-\delta, x))$, $\delta$ is used to control the distance over the surface of 3D objects and is set to 0.1 following (Park et al., 2019).

We simultaneously optimized the latent codes of training shapes $\{z_i\}_{i=1}^N$ and model parameters $\theta$ (i.e., SDF model in Figure 2). Once done training, we can obtain the latent codes for any shape via optimization: given a new shape $S_{sample}$, we initialize a latent code sampled from $p(z_{sample}) = \mathcal{N}(0, \sigma^2 I)$ and use the same objective 1 to optimize the underlying code by sampling $\mathcal{K}_{sample}$ 4-tuples describing the shape $\{c_j, s_j\}_{j=0}^{\mathcal{K}_{sample}} \sim S_{sample}$, while fixing model parameters $\theta$.

This paper demonstrates another way to obtain the latent codes by a diffusion prior. We extend the latent code notation from $z_i$ to $z_i^0$, and used the learned latent codes by the above SDF model

from our 3D datasets as the prior data distribution $\{z_1^0, \cdots, z_N^0\}$ for diffusion models. Given a sampled shape code $z^0 \sim \{z_1^0, \cdots, z_N^0\}$, the diffusion forward phase becomes $q\left(z^{0:T}\right) = q\left(z^0\right) \prod_{t=1}^{T} q\left(z^t \mid z^{t-1}\right)$, where $q\left(z^0\right)$ is the latent code distribution, and $q\left(z^t \mid z^{t-1}\right)$ is the Gaussian noise we gradually added into the latent codes. We follow (Ho et al., 2020) to model $q\left(z^t \mid z^{t-1}\right) = \mathcal{N}\left(\sqrt{1-\beta_t}z^{t-1}, \beta_t \mathbf{I}\right)$, where $\beta_1, \cdots, \beta_T$ controls the noise level added in each time step. Accordingly, we have the diffusion reverse process that recovers latent codes from standard Gaussian priors. $p_{\tilde{\theta}}\left(z^{0:T}\right) = p\left(z^T\right) \prod_{t=1}^{T} p_{\tilde{\theta}}\left(z^{t-1} \mid z^t\right)$, where $p\left(z^T\right)$ is standard Gaussian prior and $\tilde{\theta}$ is our parameterization to approximate the conditional probabilities.

Intuitively, the diffusion forward process keeps adding controlled Gaussian noise to latent codes. In the reverse process, we try to learn a model to predict the added noise at each time step, use the predicted noise to denoise, and finally generate latent codes that lies in the learned distribution. Our diffusion objective becomes $\left\| \epsilon - g_{\tilde{\theta}}\left(z^t, t\right) \right\|^2$, $\epsilon \sim \mathcal{N}(0, \mathbf{I})$, where $z^t = \sqrt{\bar{\alpha}_t}z^0 + \sqrt{1-\bar{\alpha}_t}\epsilon$, $\alpha_t = 1 - \beta_t$, $\bar{\alpha}_t = \prod_{i=1}^{t} \alpha_i$, and $\beta_t$ is the pre-defined controlled noise level at time $t$, $g_{\tilde{\theta}}$ is our diffusion model trained to predict noise trained to minimize the $\mathcal{L}_2$ loss. Once we finish training the diffusion model $g_{\tilde{\theta}}$ over all learned latent codes, we adopt Langevin dynamics for sampling latent codes from diffusion priors which could provide a conditioning input to the latent-code-based SDF model $f_\theta$ to reconstruct a new shape. The sampling formulation (Ho et al., 2020) is written as below, where $\epsilon \sim \mathcal{N}(0, \mathbf{I})$, and we do the process iteratively for $t = T, \cdots, 1$.

$$z^{t-1} = \frac{1}{\sqrt{\alpha_t}}\left(z^t - \frac{1-\alpha_t}{\sqrt{1-\bar{\alpha}_t}}g_{\tilde{\theta}}\left(z^t, t\right)\right) + \sqrt{\beta_t}\epsilon \tag{2}$$

## 3.2 TEXT-TO-SDF

We have developed a forward and reverse process to enable our diffusion model to learn the distribution of latent codes and approximate the distribution of signed distance functions. To incorporate the text condition, we maintain the posterior over the latent code given the shape 4-tuples $S$, as in the latent-code-based SDF formulation. However, we modify the diffusion objective to enable our diffusion model to learn the distribution of latent codes conditioned on text. This approach allows us to keep the SDF model focused on reconstructing the shape from latent codes while leveraging the text input.

As depicted in Figure 2, we convert the provided shape description $\mathcal{T}_i$ into an embedding and input it into our diffusion model. We accomplish this by applying BPE encoding (Sennrich et al., 2015) to transform the text into a sequence of tokens. Then, we maintain a learnable embedding for each unique token during the training process, and obtain the text embedding $\mathcal{B}_i^\phi = \text{TextEncoding}(\mathcal{T}_i)$ by concatenating the embeddings of all tokens whose text belongs to the shape description, where $\phi$ denotes the learnable parameters of token embedding (More details in Appendix F.1). Note that, we do not utilize large language models, such as BERT (Devlin et al., 2018).

Given a SDF-Text pair $\{S_i, \mathcal{T}_i\}$, we can use learned SDF model $f_\theta$ to obtain the corresponding latent-code $z_i$, and use BPE and optimized token embeddings to get text description embedding $\mathcal{B}_i^\phi$. Then, we reconstruct the diffusion forward and reverse processes: $q\left(z_i^t \mid z_i^{t-1}, \mathcal{B}_i^\phi\right)$ and $p_{\tilde{\theta}}\left(z_i^{t-1} \mid z_i^t, \mathcal{B}_i^\phi\right)$, where $\tilde{\theta}$ denotes the parameterization of the diffusion model. Our diffusion objective changes accordingly:

$$\arg\min_{\tilde{\theta}, \phi} \left\| \epsilon - g_{\tilde{\theta}}\left(z_i^t, \mathcal{B}_i^\phi, t\right) \right\|^2, \quad \epsilon \sim \mathcal{N}(0, \mathbf{I}) \tag{3}$$

The method for constructing $g_{\tilde{\theta}}\left(z^t, \mathcal{B}^\phi, t\right)$ varies across different diffusion methods. Since our diffusion target is an embedding vector lacking spatial structure (Rombach et al., 2022), we adopt an approach similar to (Ramesh et al., 2022) and Figure 3 in (Nichol et al., 2022). Specifically, we train a transformer (Vaswani et al., 2017) with casual attention masks that concatenates embeddings in a specific order: the noised latent codes $z_i^t$, the sentence embedding $\mathcal{B}_i^\phi$, the embedding for the current time step $t$, and the embeddings of the predicted noise as output $g_{\tilde{\theta}}\left(z_i^t, \mathcal{B}_i^\phi, t\right)$.

---

**Algorithm 1** CDiffSDF training and sampling.

---

**Require:** SDF-Text pairs $\{\{S_i, \mathcal{T}_i\}_{i=1}^N\}$: $S_i = \left\{(\boldsymbol{c}_j, \boldsymbol{s}_j), \boldsymbol{c}_j = (\boldsymbol{c}_j^x, \boldsymbol{c}_j^y, \boldsymbol{c}_j^z)\right\}_{j=0}^{\mathcal{K}_{train}}$, $\mathcal{T}_i$ is text;

**Require:** Initialize latent codes $\{\boldsymbol{z}_i\}_{i=1}^N$ for each shape;

**Require:** Initialize the latent-code-based SDF model $f_\theta$, the diffusion model $g_{\tilde{\theta}}$ with $\beta_t \sim$ Cosine Schedule($t$);

---

*Training SDF and Diffusion Models*
1: **repeat**
2:     $\boldsymbol{z}_i \sim \{\boldsymbol{z}_i\}_{i=1}^N$
3:     $\{\{c_j, s_j\}_{j=0}^{\mathcal{K}_{train}}\} \sim S_i$
4:     Optimize objective 1
5: **until converged**

6:     Fix $f_\theta$, set the optimized codes as $\left\{\boldsymbol{z}_i^0\right\}_{i=1}^N$, and start to train the diffusion model;
7: **repeat**
8:     $\boldsymbol{z}_i^0 \sim \left\{\boldsymbol{z}_i^0\right\}_{i=1}^N, \mathcal{T}_i \sim \{\mathcal{T}_i\}_{i=1}^N$
9:     $\mathcal{B}_i^\phi = \text{TextEncoding}(\mathcal{T}_i)$
10:     $t \sim \text{Uniform}(\{1, \ldots, T\})$
11:     Optimize objective 3
12: **until converged**

---

*Sampling*
13:     Fix $g_{\tilde{\theta}}$, $\phi$ and perform sampling 2;
14:     Obtain the generated latent code $\boldsymbol{z}_{sampled}$;
15:     Sample SDF values $\left\{f_\theta\left(\boldsymbol{z}_{sampled}, \boldsymbol{c}_j\right)\right\}_{j=1}^{\mathcal{K}_{sample}}$;
16:     Reconstruct shape to mesh or point clouds

---

### 3.3 BOOST SHAPE RECONSTRUCTION FROM LATENT CODES

This section investigates several aspects to enhance SDF model shape reconstruction ability. By incorporating all the about to propose, we reduce the hyper-parameters from 28 (Table 1 in CurrSDF (Duan et al., 2020)) to a few $\{\lambda, \boldsymbol{k}, L\}$.

**Introducing Gaussian in Training SDF**: Our main objective is to better reconstruct shapes from the latent codes generated by diffusion models. According to the sampling process formulation 2, it involves standard Gaussian noise in the iterative process and ideally converges to the samples that satisfy the learned distribution. However, in practice, we factorize both forward and reverse processes into limited discrete steps, resulting in extra Gaussian perturbation in the outcomes. Unlike text-to-2D diffusion models trained on web-scale data (Ramesh et al., 2022; Rombach et al., 2022; Saharia et al., 2022), our 3D text pair dataset has a much smaller number of samples. Fewer samples may make the model less aware of the high-dimensional data space, therefore, we assume that diffusion models trained on our limited data are less robust to Gaussian perturbations introduced during sampling than 2D diffusion models.

To verify and overcome the limitation we faced, we introduced Gaussian noise in training latent-code-based SDF to make it robust to small Gaussian perturbations on latent codes. Mathematically, we change the original objective 1 as below:

$$\underset{\theta, \{\boldsymbol{z}_i\}_{i=1}^N}{\arg\min} \sum_{i=1}^N \left( \sum_{j=1}^K \mathcal{L}\left(f_\theta\left(\boldsymbol{z}_i + \boldsymbol{\epsilon}_i, \boldsymbol{c}_j\right), s_j\right) + \omega \left\|\boldsymbol{z}_i\right\|_2^2 \right) \tag{4}$$

Where, $\boldsymbol{\epsilon}_i \sim \mathcal{N}\left(0, \bar{\boldsymbol{\sigma}}^2 \boldsymbol{I}\right)$ and $\bar{\boldsymbol{\sigma}}$ controls the Gaussian noise scale, where we empirically set to a small value closing to the noise level used in the late sampling stage. Our experiments verify the improvement of introducing Gaussian noise in training the SDF model for reconstructing from the latent codes sampled generated by the diffusion model, which validates both our assumption and the effectiveness of introducing Gaussian in training SDF models.

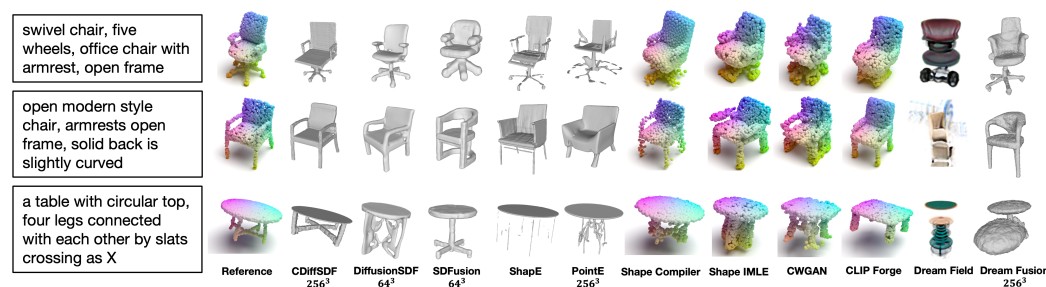

Figure 3: **Text-to-Shape results**. Many baseline methods have been unsuccessful in accurately producing structures that correspond to the detailed structural text prompts. Also, DiffusionSDF (Li et al., 2023) is limited to a pre-defined resolution of $64^3$. Compared to them, our method excels in generating corresponding structures at a high resolution of $256^3$ (or higher).

**Hard Example Mining**: In their study on curriculum learning for improving SDF model shape reconstruction performance, (Duan et al., 2020) explored curriculum learning over network depth, residual blocks, a hard example objective, and a tolerance-changing objective. Our replicated experiments using their codes found that the hard example objective brings the most improvement. Specifically, we share similar insights to (Duan et al., 2020) that detail points are often undersampled and that points in the detailed local arena may have very small loss values, making it easier for the model to predict incorrectly. To address this issue, (Duan et al., 2020) defined semi-hard and hard examples and designed the same weight term for both, where hard examples refer to the model predicting the wrong SDF sign, while semi-hard examples have a smaller SDF value than their ground-truth value but hold the correct SDF sign. However, we found that performance improved when we dropped semi-hard examples entirely and allowed the model to focus only on hard examples with the wrong SDF sign using the following weight adjustments:

$$
\begin{aligned}
&\mathcal{L}_\lambda \left( f_\theta \left( \boldsymbol{z_i} + \boldsymbol{\epsilon}_i, \boldsymbol{c}_j \right), \boldsymbol{s}_j \right) \\
&= \left( 1 - \lambda \operatorname{sgn} \left( \bar{\boldsymbol{s}}_j \right) \operatorname{sgn} \left( \bar{f}_\theta \left( \boldsymbol{z}_i + \boldsymbol{\epsilon}_i, \boldsymbol{c}_j \right) \right) \right) \mathcal{L}_{\left( f_\theta \left( \boldsymbol{z}_i + \boldsymbol{\epsilon}_i, \boldsymbol{c}_j \right), \boldsymbol{s}_j \right)}
\end{aligned}
\tag{5}
$$

where $\operatorname{sgn}$ is the sign function, $\bar{\boldsymbol{s}}_j$ is the ground-truth SDF value for point $\boldsymbol{c}_j$, and $\lambda$ controls the weight. The meaning of this weight adjustment is straightforward: pay attention to those predicted SDF values having the opposite sign to their corresponding ground-truth ones, since the points with the wrong SDF sign could have very small values with the $\mathcal{L}_1$ loss but lead to severe surface prediction errors.

While (Duan et al., 2020) addresses the issue of undersampling by implicitly encouraging the model to learn from weight changes, our paper proposes an explicit approach to solve this problem. Specifically, we introduce a sampling strategy called **Top-k** that selects the $\boldsymbol{k}$ samples with the highest loss from a total of $\mathcal{K}_{train}$ points used to compute loss $\mathcal{L}$. We then backpropagate using only those samples. In our experiments, we found that setting $\boldsymbol{k} = \frac{\mathcal{K}_{train}}{2}$ works well, which means that we optimize using only half of the sampled points in each iteration. This approach yields better results than relying on weight changes alone, as demonstrated in our experiments.

**Architecture**: Through empirical testing, we identified some architectural changes that can enhance the performance of the latent-code-based SDF model for shape reconstruction. Firstly, we adopted a technique developed in (Mildenhall et al., 2021), which involves the use of positional encoding of coordinates to consistently improve neural radiation field reconstructions. Specifically, we applied the function $\gamma(\boldsymbol{c}) = \left( \sin \left( 2^0 \pi \boldsymbol{c} \right), \cos \left( 2^0 \pi \boldsymbol{c} \right), \cdots, \sin \left( 2^{L-1} \pi \boldsymbol{c} \right), \cos \left( 2^{L-1} \pi \boldsymbol{c} \right) \right)$, where $\boldsymbol{c} \in \mathbb{R}$ represents a coordinate and $L$ is a hyperparameter that controls the length of our positional encoding. This function maps the three-dimensional coordinates to a higher-dimensional space, resulting in shape $S_i = \{ (\gamma(\boldsymbol{c}_j), \boldsymbol{s}_j) \}_{j=0}^{\mathcal{K}}$. Additionally, we observed that removing all dropout layers (Srivastava et al., 2014) and eliminating the latent connection between the first and fourth layers in the architecture of (Park et al., 2019) led to improved performance on validation sets.

(a) **Text-to-Shape comparisons.**

| Dataset | Method | IoU ↑ | FID ↓ | MMD ↓ | TMD ↑ |
|---|---|---|---|---|---|
| ShapeGlot | CWGAN | 0.098 | 121.7 | 22.46 | 0.67 |
|  | Shape IMLE | 0.153 | 87.9 | 12.37 | 1.83 |
|  | CLIP-Forge | 0.052 | 94.5 | 36.43 | 2.45 |
|  | Shape Compiler | - | - | **6.19** | 1.85 |
|  | VoxelDiffSDF | **0.207** | 76.2 | 7.35 | 2.41 |
|  | CDiffSDF | 0.196 | **73.4** | 6.42 | **2.87** |
| Text2shape | CWGAN | 0.127 | 113.2 | 10.42 | 0.79 |
|  | Shape IMLE | 0.165 | 98.4 | 6.73 | **2.34** |
|  | CLIP-Forge | 0.114 | 92.5 | 5.16 | 1.34 |
|  | Shape Compiler | - | - | 6.21 | 1.53 |
|  | VoxelDiffSDF | 0.214 | 90.2 | **4.23** | 2.25 |
|  | CDiffSDF | **0.231** | **87.3** | 4.41 | 2.17 |
| ABO | CWGAN | 0.132 | 89 | 12.74 | 0.52 |
|  | Shape IMLE | 0.196 | 86.3 | 6.43 | 1.42 |
|  | CLIP-Forge | 0.147 | 75.4 | 7.89 | 1.23 |
|  | Shape Compiler | - | - | 4.93 | 0.67 |
|  | VoxelDiffSDF | 0.235 | 59.6 | 5.21 | **1.53** |
|  | CDiffSDF | **0.253** | **56.7** | **4.71** | 1.36 |

(b) **Parameter numbers & Inference speed.**

| Methods | Enc. Params. | Gen. Params. | Speed (s) ↓ |
|---|---|---|---|
| Dream Fields | - | 149,691,777 | > 10000 |
| Dream Fusion[1] | - | 1,289,952,427 | 2432.72 |
| CWGAN | 4,183,618 | 44,101,313 | 4.24 |
| Shape IMLE | 11,149,824 | 117,534,920 | 1007.48 |
| CLIP-Forge | 5,409,665 | 18,373,120 | 1.45 |
| Shape Compiler | 26,795,008 | 36,741,120 | 6.43 |
| Point-E[2] | - | 80,804,376 | 28.17 |
| Shap-E (stf) | 443,771,357 | 315,692,032 | 16.18 |
| Shap-E (nerf) | 443,771,357 | 315,692,032 | 192.62 |
| SDFusion | 26,349,788 | 413,540,739 | 9.12 |
| VoxelDiffSDF | 13,415,497 | 465,902,920 | 8.41 |
| CDiffSDF | 3,354,502 | 3,904,640 | 1.87 |

(c) **Text-to-Shape results of pre-trained models**

| Dataset | Method | FID ↓ | MMD ↓ | TMD ↑ |
|---|---|---|---|---|
| ShapeGlot | Point-E | 104.6 | 23.27 | 2.35 |
|  | Shap-E (stf) | 92.1 | 11.45 | 2.07 |
|  | SDFusion | 78.1 | 5.87 | 2.58 |

Table 1: **Text-to-Shape methods comparison**. MMD (quality) and TMD (diversity) are multiplied by $10^3$ and $10^2$, respectively. The reported inference speed is for a single feedforward pass.

## 4 EXPERIMENTS

In this section, we evaluate CDiffSDF's performance on the Text-to-Shape task, focusing on geometry aspects. We adopt a decoder-only three-layer transformer (Vaswani et al., 2017; Ramesh et al., 2022) with a causal attention mask for our diffusion model. Following Algorithm 1, we train the latent-code-based SDF models on all the training shapes. Subsequently, we use these latent codes to train our diffusion model. For more implementation details, see Appendix F.1. We also examine the effectiveness of the proposed SDF model modifications.

**Datasets**: We utilize 3D-text pairs that offer detailed structural descriptions, excluding color details, sourced from Text2Shape (Chen et al., 2018), ShapeGlot (Achlioptas et al., 2019), ABO (Collins et al., 2021), and ShapeCompiler (Luo et al., 2023a) datasets. These datasets encompass a total of 20,355 shapes, averaging 9.47 words per description and comprising 26,776 unique words. For data examples and additional details, refer to Appendix E. Our dataset split allocates 85% for training and 15% for testing.

**Baselines**: We categorize and compare three groups of methods: (1) methods employing 3D-text pairs for model training, including Text2Shape (Chen et al., 2018), Shape IMLE (Liu et al., 2022), Shape Compiler (Luo et al., 2022), SDFusion (Cheng et al., 2023), and VoxelDiffSDF (Li et al., 2023); (2) methods such as CLIP-Forge, DreamFields, and DreamFusion (Jain et al., 2021; Sanghi et al., 2022) primarily leveraging 2D-text pre-trained models (Radford et al., 2021; Saharia et al., 2022) for 3D shape generation; and (3) robust pre-trained Text-to-3D models like Point-E (Nichol et al., 2022) and Shap-E (Jun & Nichol, 2023), trained on web-scale 3D-text pairs. For the first category, we fine-tune four methods (Chen et al., 2018; Liu et al., 2022; Luo et al., 2022; Li et al., 2023) on our datasets using their provided pre-trained models, comparing them solely in geometrical aspects (the similar protocol used in (Mittal et al., 2022)). We encountered difficulties in fine-tuning SDFusion and report its pre-trained model performance. Despite the slow testing of DreamFields and DreamFusion due to extensive sampling, we include only their qualitative results and statistics. For assessing Point-E and Shap-E, we employ the OpenAI pre-trained models to examine their competence in managing complex geometrical text prompts. Refer to Appendix F.2 for more details on baseline implementation.

**Metrics:** To assess Text-to-Shape performance, we employ a range of metrics including the render-based FID, IoU, and the point-cloud-based MMD and TMD. These metrics effectively measure the

---

[1]We utilized the dream fusion implementation based on stable-diffusion (Tang, 2022).

[2]We combined the parameters of both $base\_model$ (40,333,836) and $upsampler\_model$ (40,470,540) used in Point-E, excluding the parameters of the Point-to-SDF model.

(a) **Ablation Studies**: Evaluation results are on validation sets.

(b) **More Text-to-3D results**.

| Gaussian | Classifier-free | Clamp | IoU ↑ | FID ↓ | MMD ↓ | TMD ↑ |
|----------|-----------------|-------|-------|-------|-------|-------|
| ✗ | ✗ | ✗ | 0.142 | 109.7 | 11.38 | 1.17 |
| ✓ | ✗ | ✗ | 0.186 | 104.2 | 9.57 | 2.86 |
| ✓ | ✓ | ✗ | 0.036 | 133.8 | 20.35 | 11.27 |
| ✓ | ✓ | ✓ | 0.213 | 89.3 | 8.81 | 3.13 |

a chair with armrests, thick layers, soft

a table with slats between legs, rectangular top

Table 2: **Ablation studies & Qualitative results**. More visualizations and novel shape generation examples (with nearest neighbor retrieval (Erkoç et al., 2023)) in Appendix C.

fidelity (IoU, FID), quality (MMD), and diversity (TMD) of the generated results, as detailed below. Each model processes an input text to produce 48 normalized point clouds, each with 2,048 points and $32^3$ voxels for comparison. This voxel resolution is used because the compared method is limited to this resolution (Chen et al., 2018).

*IoU*: Intersection over Union (IoU) measures the occupancy similarity between the predicted voxel and its corresponding ground truth. It reflects the fidelity of the generated shape.

*FID*: A lower Fréchet Inception Distance (FID) (Heusel et al., 2017) indicates that the two distributions are closer, meaning that the generated shapes are of higher quality and resemble real shapes more closely. For all the compared methods except Shap-E, DreamFields, and DreamFusion, we adopted identical rendering procedures excluding color and texture.

*MMD*: Minimal Matching Distance (MMD) (Achlioptas et al., 2018) to check if the generated shape distributions are close to ground-truth shape distributions by computing the distance between the set of all our generated point clouds and the set of all ground-truth point clouds. We use Chamfer distance for each pair of point clouds in computing MMD.

*TMD*: For measuring diversity, we adopt Total Mutual Difference (TMD) (Wu et al., 2020), which computes the difference between all the generated point clouds of the same text inputs. For each generated point cloud $P_i$, we compute its average Chamfer distance $CD_{P_i}$ to other $k-1$ generated point clouds $P_{j\,j \neq i}$ and compute the average: $TMD = \text{Avg}_{i=1}^{k} CD_{P_i}$.

**Results Analysis**: Comparisons are illustrated in Table 1a and Figure 3. Both qualitative and quantitative results affirm that CDiffSDF generally secures competitive results comparable to CLIP-Forge (Sanghi et al., 2022), CWGAN (Chen et al., 2018), Shape IMLE Diversified (Liu et al., 2022), Shape Compiler (Luo et al., 2022), and VoxelDiffSDF (Li et al., 2023). Despite certain metrics and datasets where Shape IMLE, Shape Compiler, and VoxelDiffSDF surpass CDiffSDF, these models all possess significantly more parameters compared to CDiffSDF. While VoxelDiffSDF (Li et al., 2023) produces high-quality generated meshes, it is constrained to a preset resolution of $64^3$. In contrast, our method does not impose any restrictions on the resolution for output meshes. One notable drawback observed in CWGAN (Chen et al., 2018) and Shape IMLE (Liu et al., 2022) lies in the utilization of 3D convolutional layers, potentially leading to overfit the training data distribution.

A comparison with Point-E and Shap-E reveals that a lightweight framework can outperform these larger models with training over limited high-quality 3D-text data in the specific geometrical-text-guided 3D shape generation task. Despite the proven success of employing CLIP in 3D shape generation with text prompts containing straightforward structure descriptions (Sanghi et al., 2022; Jain et al., 2021; Poole et al., 2022), both DreamFields and DreamFusion fall short in generating detailed structures that correspond to the text prompts.

**Model Size and Inference Speed**: we investigate the number of trainable parameters and inference speed for the methods we compare. Most of these methods consist of two stages: the first stage encodes shapes and texts into embeddings, while the second stage learns the target embeddings by executing the learned generative model. The exception to this is DreamFields and DreamFusion, which is primarily a generative model composed of a small NeRF and a Image-Text model (using ViT-B/16 or stable-diffusion). Additionally, Point-E works directly on point clouds and does not require an encoder. Since some of the compared methods employ large-scale pre-trained language models, it is not meaningful to compare the parameters of text encoders. Therefore, we do not include the text encoder parameters in our comparison, including Bert used in Liu et al. (2022),

(a) **SDF reconstruction quantitative comparisons.**

(b) **SDF reconstruction qualitative comparisons.**

| Method | Chair | Sofa | Table | Lamp | Plane | Avg. ↓ |
|---|---|---|---|---|---|---|
| AtlasNet-Sph | 0.752 | 0.445 | 0.725 | 2.381 | 0.188 | 0.730 |
| AtlasNet-25 | 0.368 | 0.411 | 0.328 | 1.182 | 0.216 | 0.391 |
| DeepSDF | 0.243 | 0.117 | 0.424 | 0.776 | 0.143 | 0.319 |
| CurrSDF | 0.153 | 0.099 | 0.302 | 0.465 | 0.072 | 0.218 |
| CDiffSDF | **0.109** | **0.087** | **0.273** | **0.422** | **0.061** | **0.190** |

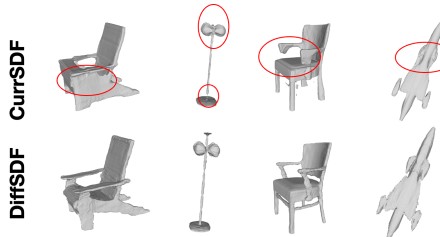

Table 3: **Shape reconstruction from latent codes comparisons**. Numbers are mean CD with $\times 10^3$. Our modifications lead to an improvement of 12.8%.

CLIP used in Sanghi et al. (2022), spaCy (Honnibal et al., 2020) used in Chen et al. (2018), CLIP used in Li et al. (2023), and BPE + learnable text embeddings used in (Luo et al., 2022) and our method. We measure the average inference time for one test batch using a single A5000. For lightweight methods, including ours, the time taken to generate 48 samples is recorded (one test batch). In contrast, for heavier or slower models like ShapE and DreamFusion, the time is calculated for producing just one sample (also one test batch).

Table 1b highlights the results, underscoring the benefits of employing our proposed method in the compact latent embedding space for the diffusion model. Our approach minimizes both model size and GPU memory usage, enhancing feasibility for Text-to-3D operations under computation-sensitive conditions. Appendix B outlines an alternative, utilizing the diffusion model over a voxel-like SDF representation (Shen et al., 2021). This alternative, however, proves to be slower and less efficient in performance compared to our latent-code-based solution. Additionally, our usage of 80 sampling steps in the diffusion model bolsters quick inference time, with potential for further enhancement as diffusion model techniques advance.

**Ablation Studies**: As shown in Sec. 3.3, we proposed to add Gaussian noise in training DeepSDF to make the latent-code-based SDF model robust to potential Gaussian noise introduced by the sampling process 2. We verify the effectiveness of this module in Table 2a, where we also examine the effectiveness of introducing classifier-free guidance (Ho & Salimans, 2022) and $\mathcal{L}_2$ norm clamping for the diffused embedding at each denoising iteration: $v = \frac{v}{\max(\|v\|_2, \epsilon)}$, where $v$ is the intermediate embedding. This operation is similar to clamp images in the range of $[-1, 1]$. Results are reported in the validation dataset, for which we held out 10% of training data as validation data. Results show introducing Gaussian in training the SDF model brings improvements. Performing classifier-free guidance alone causes drastic performance drops while combining with the clamp technique can lead to small benefits. This phenomenon is consistent with the finding in (Saharia et al., 2022). The very high TMD score achieved by only using Classifier-free guidance is caused by the generated shapes containing many random fragments and unrelated structures. Therefore, it is important to measure Text-to-Shape performance from multiple angles.

**Shape Reconstruction from Latent Codes**: We also investigate the impact of the design choice, proposed in Section 3.3, on the shape reconstruction performance of the latent-code-based SDF model. Here we adopt $L = 10$ for positional encoding and $\mathbf{k} = 0.5 \times \mathcal{K}_{train}$ for Top-k loss computing, where $\mathcal{K}_{train} = 16384$ following (Park et al., 2019). More ablation around $L$ and $\mathbf{k}$ are included in Appendix D. Results are shown in Table 3a and Figure 3b which demonstrate the effectiveness of the proposed architecture adjustment and online hard example mining mechanism.

## 5 CONCLUSION

In this work, we introduce CDiffSDF, a lightweight Text-to-Shape framework that employs SDF representation and applies diffusion across the latent space. This innovative approach empowers CDiffSDF to efficiently generate intricate 3D shapes, while maintaining a more compact model size relative to other Text-to-Shape methods. Our comprehensive investigation into various design choices for the latent-code-based SDF model encompasses aspects such as positional encoding, hard example mining, and the incorporation of Gaussian noise during training. The harmonious integration of these elements further augments CDiffSDF's capability to adeptly produce 3D shapes.

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

## A  LIMITATIONS AND FUTURE WORK

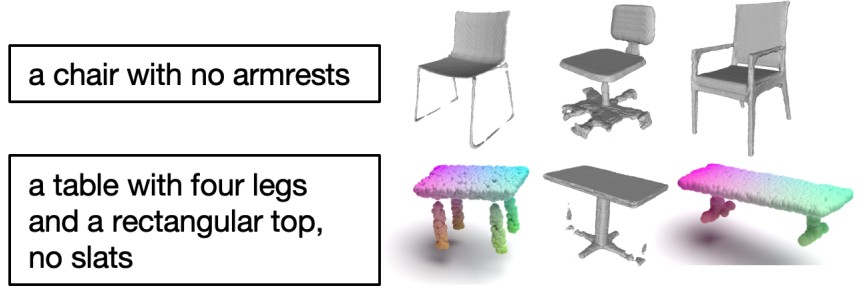

Figure 4: The failure cases generated by CDiffSDF.

Our framework predominantly yields two categories of generation failures. The first category entails the inability to accurately reconstruct the overall shape, typically manifesting as disjointed refactored components and artifacts, illustrated in the 1st and 2nd columns of Figure 4. The second category involves a mismatch between the reconstructed shapes and the input text prompt, highlighted in the 3rd column.

Mitigation of the first type of failure can potentially be achieved with an augmentation of training data. This enhancement allows the diffusion model to more precisely approximate the latent code distribution and enables the latent code-based SDF model to more robustly reconstruct shapes from the latent codes. Addressing the second category of failures necessitates the development of advanced techniques to enhance the attention of diffusion models to conditions. An increase in shape-text pairs could also contribute to the alleviation of this issue.

Echoing the requirements of (Chen et al., 2018; Liu et al., 2022; Li et al., 2023), CDiffSDF necessitates sizable 3D-text pairs for model training. This demand underscores the need for the community to meticulously annotate 3D shapes through crowdsourcing to amass larger datasets, albeit a costly endeavor. A recent advancement involves utilizing pre-trained captioning models and Large Language Models (LLM) to annotate expansive 3D datasets (Luo et al., 2023b; Deitke et al., 2023). Another promising approach entails the recovery of 3D shapes (Gkioxari et al., 2022; Qian et al., 2022) from 2D/Video-Text pairs, subsequently yielding 3D-text pairs.

Moreover, here is no assured guarantee that the proposed framework can seamlessly scale up with web-scale 3D-text pairs given the adoption of a limited-size encoder and diffusion model. Besides, a crucial future development involves the expansion of our framework to incorporate texture, while maintaining its lightweight characteristics.

## B  VOXEL-BASED SDF TEXT-TO-SHAPE

As mentioned in Section 4, we have studied voxel-like SDF representation (Shen et al., 2021) before moving into the latent-code-based SDF formulation. This section shares some of our discoveries. For the voxel-like SDF representation, we refer to a voxel $32 \times 32 \times 32$, where each cell stores the corresponding SDF value, as shown in the left of Figure 5. We can reconstruct the mesh based on the voxel representation via marching cubes as shown in the right of Figure 5.

Accordingly, we used a diffusion model that can consume the voxel-like SDF representation, where we adopt a U-Net structure with three-levels of downsampling and upsampling, 3D convolutional layers, and Group Normalization layers (Wu & He, 2018). We follow the same training setting as we used in our latent-code-based SDF diffusion models, listing in Appendix F.1.

However, our results show the voxel-like SDF + 3D UNet diffusion models failed to capture the 3D object distribution, as shown in Figure 6. The diffusion inference of the voxel-like SDF representation costs 110.38 seconds for sampling a 48 batch which significantly slower than the latent-code-based SDF solution which cost 15.21 seconds. Therefore, we do not further play with the voxel-based solution and delve deep in the latent-code-based SDF representation.

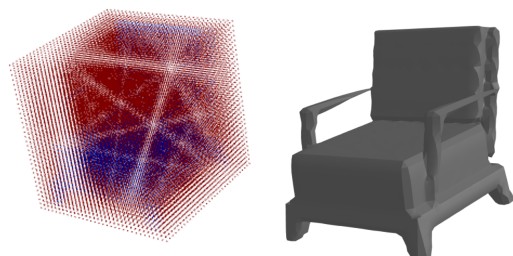

Figure 5: An instance of voxel-like SDF representation. Blue points mean inside the shape, and red points mean outside the shape. The right mesh is the reconstruction mesh on the left voxel SDF. Visualization is achieved via Pyrender with 'use_raymond_lighting == True'.

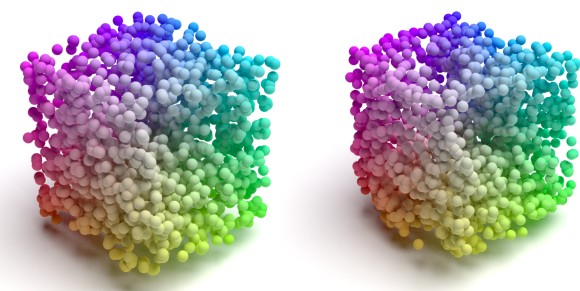

Figure 6: The Voxel-like SDF representation + 3D UNet diffusion model failed to capture the distribution of 3D shapes.

## C   MORE RESULTS

This section provides more results generated by *CDiffSDF*.

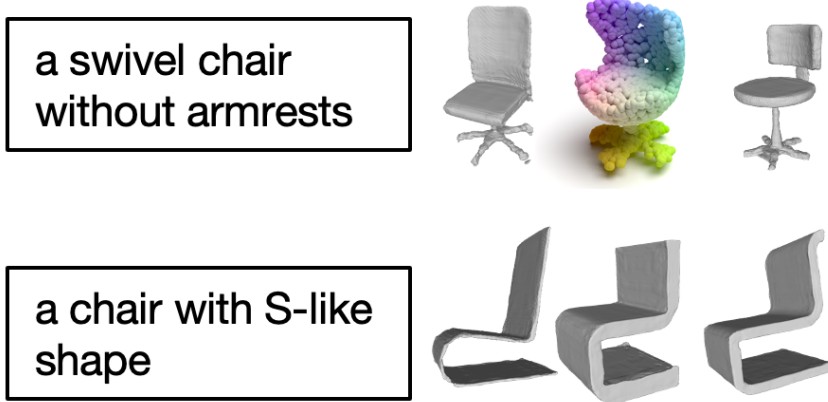

Figure 7: More results generated by CDiffSDF. The second row shows many different details contained in shapes that reflect the same text prompt.

Furthermore, we check whether CDiffSDF shows combinatorial generalizability that can generate new shape structures by composing known structures. We check this by looking at the nearest neighbors (Chamfer Distance) of the generated shape in our training data. Results are listed in Figure 10 and show our framework generates novel structures beyond training shapes to some extent.

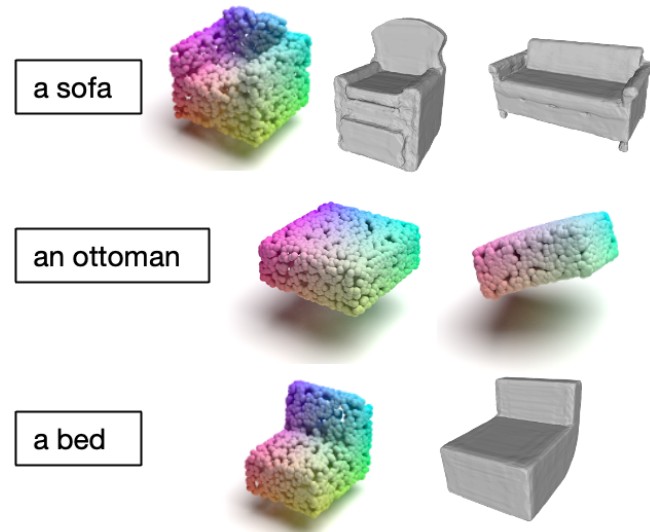

Figure 8: More results generated by CDiffSDF.

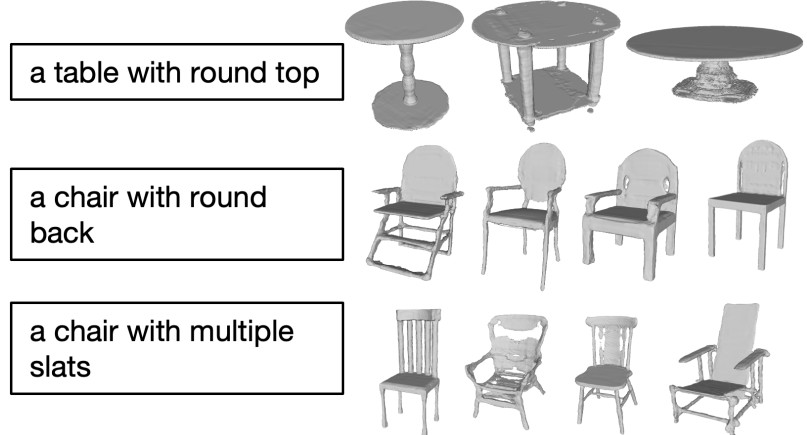

Figure 9: More results generated by CDiffSDF.

## D    SHAPE RECONSTRUCTION FROM LATENT CODES

For our latent-code-based SDF framework (Figure 11), we adopt the same 8 fully connected layers as (Park et al., 2019; Duan et al., 2020), but no latent connections, no Dropout layers, and the input dimensions change from $259 = 256 + 3$ to $296 = 256 + 30$, where we adopt $L = 10$ to positional encode three-dimensional coordinates. Following (Park et al., 2019), we sampled 500,000 points from each shape before training, and sampled $\mathcal{K}_{train} = 16,384$ points during each training optimization, and $\mathcal{K}_{sample} = 8,000$ in each inference optimization. We trained all the SDF models for 2,000 epochs with batch size 96, learning rate $1e-3$ for the latent codes optimization, and learning rate $5e-4$ for the model optimization. The code regularization term is set to $\omega = 1e-4$. We add Gaussian noise with mean $0$ and std $0.05$ during training, as mentioned in Section 3.3. Also, unlike (Duan et al., 2020), which adjusted $\lambda$ by manually setting a series of intervals, we set up a continuous sigmoid-like schedule to gradually increase $\lambda$ from 0 to 0.5: $0.5 * \left(1 - 1/1 + \exp\left(\left(e - \frac{\varepsilon}{2}\right) / \frac{\varepsilon}{20}\right)\right)$, where $exp$ stands for exponential, $e$ denotes the current epoch number, and $\varepsilon$ denotes the total training epochs.

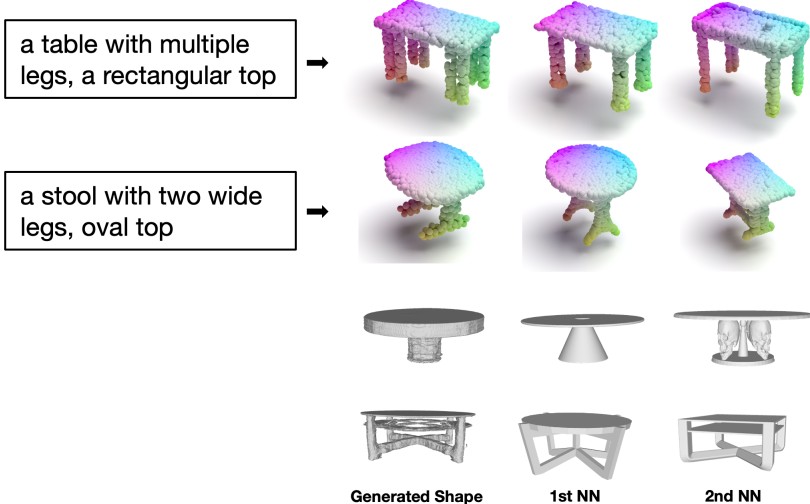

Figure 10: We inspect the top-2 nearest neighbors of the generated shapes in training datasets to check if CDiffSDF can compose known structures to generate new ones. The first two rows are text-to-shape generation, and the bottom two rows are unconditional generation.

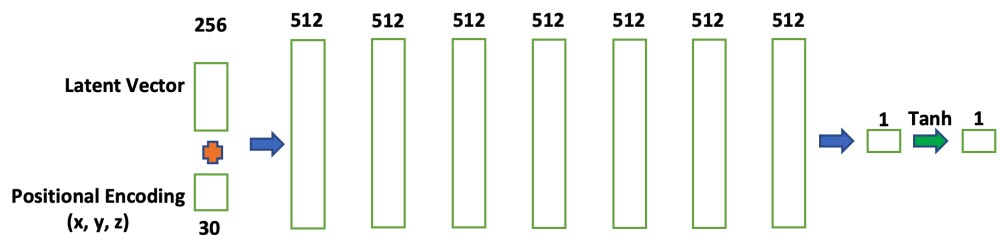

Figure 11: Network architecture illustration for our latent-code-based SDF model. All layers are connected via fully connected layers except the green arrow represents the Tanh activation function.

The rightmost column of Table 4 shows the results where we leave a subset of training shapes as validation dataset and tested the average Chamfer Distance between the reconstructed shape and its corresponding ground-truth one.

| | PosEncoding | Latent | Dropout | Top-k | Weight | Score ↓ |
|---|---|---|---|---|---|---|
| DeepSDF | ✗ | ✗ | ✗ | ✗ | ✗ | 0.175 |
| - | ✗ | ✗ | ✗ | ✓ | ✓ | 0.137 |
| - | ✓ | ✗ | ✗ | ✗ | ✗ | 0.152 |
| - | ✓ | ✓ | ✗ | ✗ | ✗ | 0.15 |
| - | ✓ | ✓ | ✓ | ✗ | ✗ | 0.143 |
| - | ✓ | ✓ | ✓ | ✓ | ✗ | 0.119 |
| Ours | ✓ | ✓ | ✓ | ✓ | ✓ | 0.113 |

Table 4: **SDF model ablation studies**: numbers are mean CD with $\times 10^3$, tested in normalized validation sets.

Besides, we conducted further ablation studies on the selection of $L$ which controls the length of our positional encoding, and $k$ which controls the hard example mining magnitude. In our main experiments, we adopt $L = 10$ and $k = 0.5$.

|              | Score ↓ |
|--------------|---------|
| Ours-$k = 0.3$ | 0.117 |
| Ours-$k = 0.7$ | 0.126 |
| Ours-$L = 1$   | 0.128 |
| Ours-$L = 30$  | 0.119 |
| Ours           | 0.113 |

Table 5: **SDF model more ablation studies:** numbers are mean CD with $\times 10^3$, tested in normalized validation sets.

# E    DATASET

Here we provide some examples around the dataset (Luo et al., 2022) we used. It consist of shapes from Text2Shape (Chen et al., 2018), ABO (Collins et al., 2021), and ShapeGlot (Achlioptas et al., 2019) datasets. All the shapes in the dataset are furniture, and the main parts of them are chairs and tables, which contain a lot of structure variations.

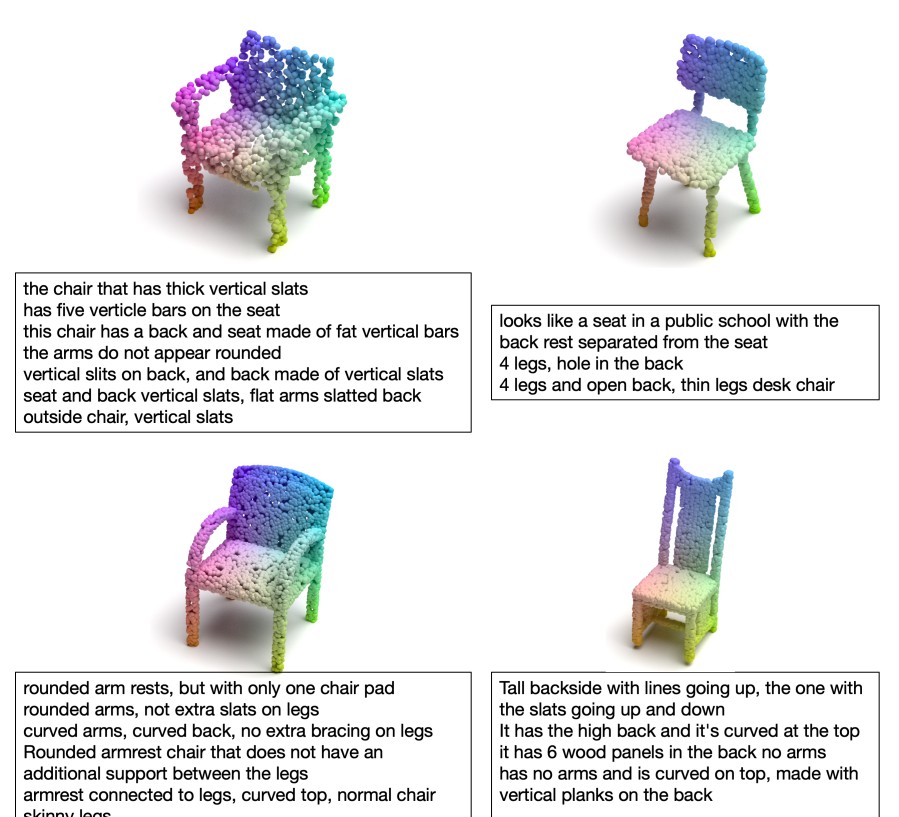

Figure 12: Random examples of shapes from ShapeGlot (Achlioptas et al., 2019). One description per sentence.

# F    IMPLEMENTATION DETAILS

This section lists model and training details around the proposed framework and the compared baselines. We also list examples of the adopted data, (Shape, Structure-Related Text), to provide some sense for our problem setting.

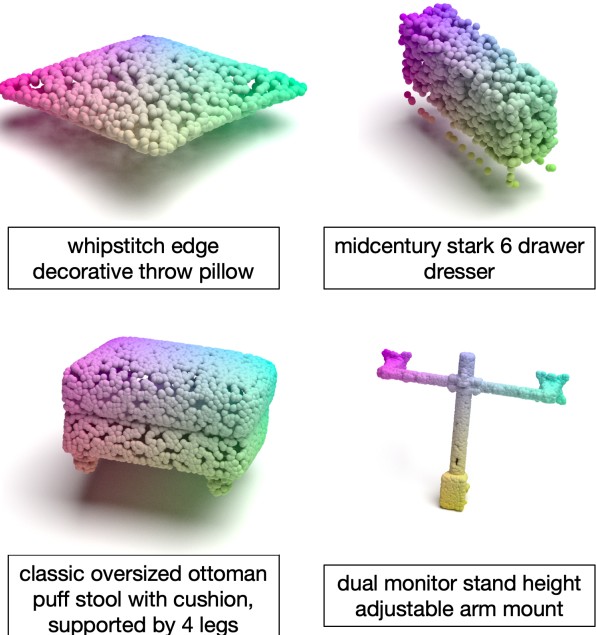

Figure 13: Random examples of shapes from ABO (Collins et al., 2021). One description per sentence.

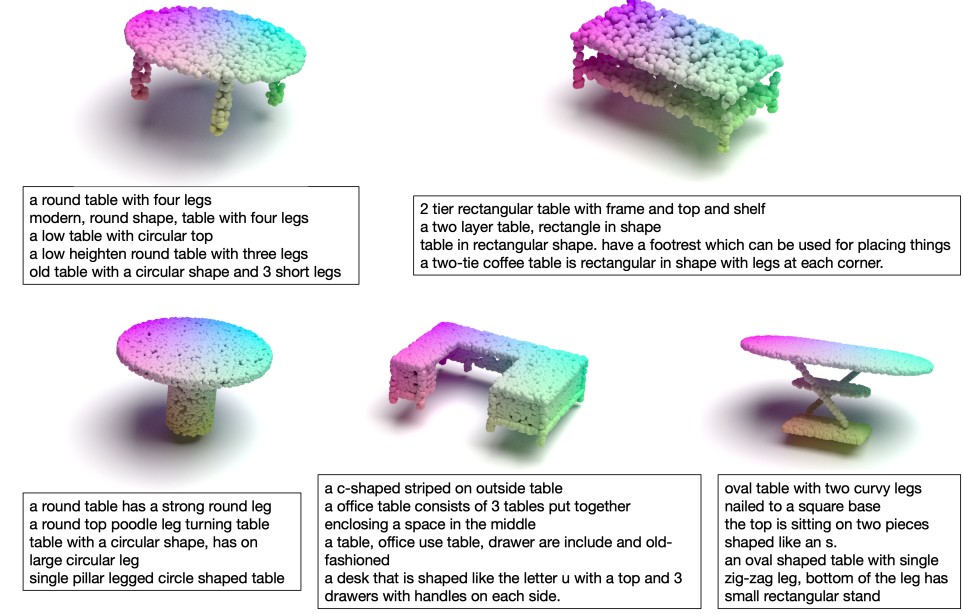

Figure 14: Random examples of shapes from Text2Shape (Chen et al., 2018). One description per sentence.

## F.1 CDiffSDF

We adopt a decoder-only transformer (Vaswani et al., 2017) with casual attention (Child et al., 2019; Ramesh et al., 2022) as the diffusion model. The Transformer has depth 3, MLP wide 64, and 8 attention heads. We train the diffusion model with batch size 256, learning rate starting from $1e-3$ following by a cosine annealing schedule, the ADAM optimizer with default parameters, and a total of 10,000 epochs. We used a linear schedule to regulate $\beta_t$ and compute $\alpha_t$ and $\bar{\alpha}_t$ accordingly.

We set $\beta_{t_0} = 0.004$ and $\beta_{t_{1000}} = 0.00002$ decided by the noise level used in training latent-code-based SDF model. For classifier-free guidance (Ho & Salimans, 2022), we used drop ratio $0.1$. The diffusion training and sampling objective are listed in Section 3.2.

For visualizations used in this paper, we adopt Misuba (Jakob et al., 2022) to render point clouds and meshlab (Cignoni et al., 2008) to render meshes.

For other types of conditions, the general approach mirrors that of Text-to-3D conversion. Initially, input conditions are converted into embeddings, denoted as $\mathcal{B}_i$. These embeddings act as conditions in our learned diffusion model to predict a latent code, $z_i$. In the case of image-conditioned scenarios, a frozen CLIP image encoder is utilized. For class-conditioned scenarios, one-hot encoding is employed for all classes. In unconditional situations, the condition input is omitted, directly applying the diffusion model to the latent codes of 3D objects.

## F.2 BASELINE DETAILS

We compared with various Text-to-Shape baselines to examine the performance of our proposed *CD-iffSDF*. For each baseline, we generally follow the default configurations provided by their Github, including learning rates, batch size, training epochs, and other hyper-parameters. The major modification we made is the data part, where different methods adopt different ways to process 3D objects and text. We list some of our modifications below:

- **Shape Compiler (Luo et al., 2022)**: We compare with Shape Compiler Limited listed in (Luo et al., 2022), due to we only use 3D-Text data and perform the Text-to-Shape task solely.

- **CLIP-Forge (Sanghi et al., 2022) and Dream Field (Jain et al., 2021)**: We follow their papers and use *clip.tokenize* and *clip.encode_text* to encode text. We used the pre-trained model provided in CLIP-Forge's Github to generate multiple shapes given one text prompt. We adopt the Pytorch implementation of Dream Field (Jain et al., 2021) for qualitative comparisons as shown in Figure 3. We do not adopt the provided codes in (Jain et al., 2021) Github because it takes more than eight days to process one single text prompt with our 4 GPUs of 2,048 Gb memory. Although, the Pytorch implementation still costs more than 10,000 seconds for a text prompt. Therefore, it is impractical to test Dream Feild in our test set, which consists of 15,000 text-shape pairs.

- **Shape IMLE (Liu et al., 2022)**: we process all text with Bert (Devlin et al., 2018) follow their codes (Liu et al., 2022). We adopt *BertTokenizer* and of the pre-trained flag "bert-base-uncased" to tokenize the text and then obtain the embedding by the Bert model with the same pre-trained flag.

- **CWGAN (Chen et al., 2018)**: we follow the data templates provided in (Chen et al., 2018) to process our data. We process each data entry as in (Chen et al., 2018), including *caption_tuples*, *caption_matches*, *vocab_size*, *max_caption_length*, and *dataset_size*. For the text tokenizer, we used spaCy (Honnibal et al., 2020) as mentioned in (Chen et al., 2018). (Chen et al., 2018) set the token number in a linear increase rule based on their data order, while we set our token number based on our data order. We abandoned descriptions with more than 96 tokens as (Chen et al., 2018) did.

- **CurrSDF (Duan et al., 2020) and DeepSDF (Duan et al., 2020)**: we follow their GitHub codes to train and implement without any code changes. The differences between CurrSDF (Duan et al., 2020) and DeepSDF (Park et al., 2019) are summarized in Table 1 of (Duan et al., 2020), where they manually set up a lot of stages to gradually change network structures and loss weights.

- **DreamFusion (Poole et al., 2022)**: we adopted the implementation provided by https://github.com/ashawkey/stable-dreamfusion. We use the plain NeRF implementation instead of Instant-NGP.

- **VoxelDiffSDF (Li et al., 2023)**: we directly follow the instructions provided by their office code release.

- **Point-E (Nichol et al., 2022) Shap-E (Jun & Nichol, 2023)**: we used the pre-trained model provided by OpenAI Githubs. For Shap-E, we reconstruct the output mesh via STF mode.

