# OpenReview forum: "Compact Text-to-SDF via Latent Modeling"
_ICLR.cc/2024/Conference — Submitted to ICLR 2024_

### Official Review · Reviewer_jSYM · 2023-10-13

**Soundness:** 3 good
**Presentation:** 3 good
**Contribution:** 3 good
**Rating:** 6
**Confidence:** 4

**Summary:**

This paper introduces a novel methodology for Text-to-Shape synthesis, leveraging latent SDF in conjunction with the diffusion process to facilitate the generation of object shapes. The proposed method allows for generation conditioned on a diverse range of inputs, including text, object categories, and images. To enhance its generative capabilities, some attempts including Gaussian noise and hard example mining are introduced, and experiments verify their efficacy.

**Strengths:**

1.	This paper leverages latent SDF as the 3D shape representation, and the model is lightweight since the diffusion denoising process is conducted within the latent space.
2.	The method allows flexible inputs and generates reasonable results for both text and image.
3.  The paper is well-written and experiments show good qualitative results compared to baselines.

**Weaknesses:**

1.	An SDF-based representation can only represent the geometry of the object without appearance, which limits the usage of the generated shape.
2.	The categories of generated objects demonstrated in the experiment are very limited. Is it possible to generate more flexible objects (for example, hamburger)?
3.	The results of image-conditioned generation in Fig.1 seem not so similar to the input image.

**Questions:**

Text-to-shape models trained with 3D-text pairs seem to highly rely on the training data, while 3D data is much more difficult to obtain than 2D images. Will this lead to a lower diversity of generated results than methods based on text-to-image?

---

> ### Author Response · Authors · 2023-11-22
> **Author Response**
>
> We greatly thank Reviewer jSYM for the valuable comments and suggestions.
>
> First and foremost, we would like to address your concerns regarding the diversity of generated results in text-to-3D methods. It's true that text-to-3D methods rely on 3D-text pairs, which can be more challenging to acquire compared to image-text pairs, potentially impacting the diversity of generated outputs. However, recent advancements in 3D-text datasets, such as Objaverse [1] and Cap3D [2], which provide access to 660k 3D-text pairs, have the potential to alleviate this issue. Furthermore, the situation could significantly improve with increased computational resources to generate 3D-text pairs on Objaverse-XL [3], a dataset containing over 10 million objects. Although this dataset size is still smaller than that of image-text datasets, it's worth noting that with 3D supervision available, we can guide the model to learn intricate 3D meshes. **This is exemplified by our work, where a diffusion model demonstrated the ability to reconstruct detailed geometry with as few as ~10k 3D-text pairs, relying on direct 3D supervision.** Furthermore, our method boasts significantly fewer parameters and achieves faster processing speeds compared to existing image-to-3D approaches.
>
> [1]Objaverse: A universe of annotated 3d objects, CVPR23
>
> [2]Scalable 3D Captioning with Pretrained Models, NeurIPS23
>
> [3]Objaverse-xl: A universe of 10m+ 3d objects, NeurIPS23
>
> ---
> > The categories of generated objects demonstrated in the experiment are very limited. Is it possible to generate more flexible objects (for example, hamburger)?
>
> We seriously consider the point you mentioned in the rebuttal. To reconstruct a hamburger, we train our model on a small subset of Objaverse-Cap3D pairs [1,2]. Please refer to: https://drive.google.com/file/d/1CnzMd5MOngOxYHo_27suHdQCmDKOV_O5/view?usp=sharing, which showcases a variety of hamburgers, each with different numbers and types, demonstrating our model's ability to generalize compositionally (to some levels).
>
> In the submission, we provided more results at Appendix C. Additionally, we used the trained model in the submission to generate more results from diverse furniture categories. Please refer to https://drive.google.com/file/d/1wM9mHzUehW_FYB8ITVUZaB8Byx8JI16O/view?usp=sharing.
>
>
> ---
> > An SDF-based representation can only represent the geometry of the object without appearance, which limits the usage of the generated shape.
>
> We acknowledge this limitation. It's important to note that the primary focus of this paper is on lightweight 3D geometry generation. The current prevalence of SDF-based representations is primarily associated with detailed 3D geometry. We have the option to incorporate texture onto the 3D mesh, drawing from existing texture synthesis research [4] or utilizing texture generation products [5].
>
> [4] Text2Tex: Text-driven Texture Synthesis via Diffusion Models, 2023
>
> [5] MeshyAI, https://twitter.com/jimmygunawanapp/status/1724969804040389036
>
>
> ---
> > The results of image-conditioned generation in Fig.1 seem not so similar to the input image.
>
> We currently have very limited image-3D pairs on the Pix3D dataset. This situation could be alleviated by using Objaverse dataset. Please allow us to further improve this point later due to computation constraints.

---

### Official Review · Reviewer_qQcJ · 2023-10-31

**Soundness:** 3 good
**Presentation:** 3 good
**Contribution:** 3 good
**Rating:** 8
**Confidence:** 3

**Summary:**

The paper suggests CDiffSDF, a model that generates a 3d shape conditioned on text. It merges latent diffusion model to process text and generate 3d latents and SDF decoder to generate the final 3d shape unlike many previous approaches that use intermediate 2d generative model instead. Hence it requires paired 3d-text dataset. The authors also suggest to introduce Gaussian noise into SDF training procedure. CDiffSDF seems to get impressive results on a few benchmarks while having noticeably fewer parameters compared to many baselines. Being an implicit 3d model it benefits from higher resolution than voxel-based models.

**Strengths:**

Related work is quite extensive yet concise. The method looks quite original to me. The main ideas are described in lots of detail. The experiments show improvement over SOTA on a few datasets as well as ablations for several model variants. Inference time benchmarks also show the proposed model is pretty fast (as it has relatively few parameters). The authors also used a few well-known improvements over standard SDF architecture. Overall I find it a pretty good paper with a coherent story.

**Weaknesses:**

Even though I reread the paper multiple times, I still find it hard to fully understand how exactly the model is constructed. High level understanding is easy to get though. Open source code would be useful for the future research. Improvements in clarity of the paper are highly encouraged.

**Questions:**

I suggest to add more quality results. 3D papers tend to show more images and that is important.

---

> ### Author Response · Authors · 2023-11-22
> **Author Response**
>
> We greatly thank Reviewer qQcJ for the comments and suggestions. We appreciate the time you spent on the paper. Below we address your concerns.
>
> ---
> > Even though I reread the paper multiple times, I still find it hard to fully understand how exactly the model is constructed. High level understanding is easy to get though. Open source code would be useful for the future research. Improvements in clarity of the paper are highly encouraged.
>
> Thank you for your suggestion, we plan to incorporate the below updates into the revision upon your check.
>
> To provide a clearer understanding of our proposed method, we have created a more comprehensive illustration: https://drive.google.com/file/d/11yPZnGb6W8qRb5Q6Gf7Pem7ohUK5zZk4/view?usp=sharing. Also, our main model code is uploaded here, [train_CDiffSDF.py](https://drive.google.com/drive/folders/1m-qJEbHGVqB-rvp3WjeMkUnuxfY2RHxb?usp=sharing)).
>
> Thank you for your suggestion. We will open-source our codes to facilitate related research in our community.
>
> ---
> > I suggest to add more quality results. 3D papers tend to show more images and that is important.
>
> Thanks for the suggestion! In the submission, we provided more results at Appendix C. During the rebuttal, we  generated more results from the our model (trained in the submission) and trained our Compact-DiffSDF on a small subset of Objaverse [1] and Cap3D [2] to obtain more results from various categories. You can please refer to: https://drive.google.com/file/d/1-GYKMrnuqFQD3J1XkTJt8j0zkxbyF5KB/view?usp=sharing
>
> [1]Objaverse: A universe of annotated 3d objects, CVPR23
>
> [2]Scalable 3D Captioning with Pretrained Models, NeurIPS23

---

> > ### Comment · Reviewer_qQcJ · 2023-11-22
> >
> > Thank you, I appreciate your response. The new figure is a lot easier to understand. I am glad to see the code and additional visual results.

---

### Official Review · Reviewer_jMCT · 2023-10-31

**Soundness:** 3 good
**Presentation:** 3 good
**Contribution:** 2 fair
**Rating:** 3
**Confidence:** 4

**Summary:**

The paper presents CDiffSDF, a text-to-shape model aimed at generating 3D shapes based on textual inputs. By leveraging latent-code-based signed distance functions (i.e., DeepSDF), the model achieves relatively high-resolution shape generation while being lightweight. The paper 1) introduces Gaussian noise perturbations and 2) incorporates hard example mining during the SDF training phase to improve the generation capability. Experimental validations are provided to showcase the model's efficacy.

**Strengths:**

(+) The paper addresses an important and active research problem: text-to-3D shape generation.

(+) The paper provides a clear and well-structured explanation of the main methodology.

(+) The paper studies a less explored perspective on text-to-3D synthesis, leveraging DeepSDF as the 3D shape representation.

**Weaknesses:**

(-) Limited technical novelty: The proposed method follows a two-stage approach: Firstly, fitting the DeepSDF latent code and subsequently training conditional diffusion models in the latent space. This latent diffusion paradigm has been standard and extensively studied. The novelty of the paper appears more in the introduction and improvement of minor techniques, like 1) fitting DeepSDF with Gaussian perturbations, and 2) incorporating hard example mining, inspired by Curriculum DeepSDF. Overall, there's a noticeable lack of fresh technical insights.

(-) Recent state-of-the-art methods, such as DiffRF (CVPR 2023), 3DShape2Vecset (SIGGRAPH 2023), and Shap-E (arXiv), have also leveraged the idea of performing diffusion in the latent space of NeRF/SDF/occupancy. This paper lacks an in-depth comparison and discussion concerning these methods.

(-) The ablation study seems to miss experiments that demonstrate the role of hard example mining.

(-) The claim on page 8 regarding the lightweight framework outperforming larger models like Shap-E and Point-E ("A comparison with Point-E and Shap-E reveals that a lightweight framework can outperform these larger models with training over limited high-quality 3D-text data in the specific geometrical-text- guided 3D shape generation task.") seems unsubstantiated. To strengthen this claim, the paper should consider re-training or fine-tuning Shap-E/Point-E on the same dataset and comparing results.

**Questions:**

- In page 3, Section 3.2, the paper devises its method for obtaining text embeddings. Why not utilize pre-trained language or multimodal models?
- How does the proposed model perform when subjected to "out-of-distribution" or more creatively constructed text prompts?

---

> ### Author Response · Authors · 2023-11-21
> **Author Response (part 1)**
>
> We sincerely appreciate the detailed reviews and suggestions by Reviewer jMCT. We had anticipated many of the concerns you raised and have addressed them in our response. Following your advice, we have also carried out additional quantitative and qualitative analyses.
>
> The below is our response with our thoughts and more comprehensive results. We welcome any further thoughts you may have. Once we’ve reviewed these modifications with you, we will integrate them into the final revision.
>
> ---
> > In page 3, Section 3.2, the paper devises its method for obtaining text embeddings. Why not utilize pre-trained language or multimodal models?
>
> For the sake of compactness, we did not leverage pre-trained language/multimodal models. We employed [the CLIP model](https://github.com/openai/shap-e/blob/main/shap_e/models/generation/pretrained_clip.py), specifically ViT-L/14, as used in Shap-E, to obtain text embeddings. It improved our performance over ShapeGlot by 5.2 on FID and 0.71 on MMD. However, this improvement results in a significant increase in parameters as shown in the below table.
>
> ------------------------
> |                                                              | total       | clip.model vs. text_emd | clip_embed vs. text_pos_emb |
> | ------------------------------------------------------------ | ----------- | ----------------------- | --------------------------- |
> | [CLIP Model](https://github.com/openai/shap-e/blob/main/shap_e/models/generation/pretrained_clip.py) | 427,813,377 | 427,616,513             | 196,864                     |
> | Ours                                                         | 12,845,056  | 12,713,984              | 131,072                     |
> -------------------------
>
> The specific implementation of our current text-embedding modules and the one use CLIP are listed below. Based on the above parameter-Table, we can tell our current implementation has $0.03 \times$ parameters of the CLIP model (`12845056/427813377=0.0300249`).
>
> Thank you for raising this up, and we are happy to include this discussion in the revision.
>
>
> ```
> ## our current text modeling
> ## text are tokenized via BPE (no parameters)
> self.dim = 256
> self.text_seq_len = 512
> self.num_text_tokens = 49664 ##vocabulary size
> self.text_emb = nn.Embedding(self.num_text_tokens, self.dim)
> self.text_pos_emb = nn.Embedding(self.text_seq_len, self.dim)
>
> ## CLIP modeling (https://github.com/openai/shap-e/blob/main/shap_e/models/generation/pretrained_clip.py)
> self.clip = FrozenImageCLIP(torch.device('cuda' if torch.cuda.is_available() else 'cpu'))
> self.clip_embed = nn.Linear(
>     self.clip.feature_dim, self.dim,
> )
> ```
>
>
> ---
> > How does the proposed model perform when subjected to "out-of-distribution" or more creatively constructed text prompts?
>
> In our submission, we have included the top-2 nearest neighbor results in Appendix Figure 10, specifically for cases where the text prompts fall outside the training distribution. However, it's worth noting that for highly imaginative text prompts, such as one describing `a table with 100 legs`, the current model may struggle to generate accurate responses.
>
> ---
> > Limited technical novelty: The proposed method follows a two-stage approach: Firstly, fitting the DeepSDF latent code and subsequently training conditional diffusion models in the latent space. This latent diffusion paradigm has been standard and extensively studied. The novelty of the paper appears more in the introduction and improvement of minor techniques, like 1) fitting DeepSDF with Gaussian perturbations, and 2) incorporating hard example mining, inspired by Curriculum DeepSDF. Overall, there's a noticeable lack of fresh technical insights.
>
> We would like to point out that the introduction of Gaussian perturbation in the latent SDF modeling and diffusion model has not been previously explored. Also, Curriculum DeepSDF did not investigate a hard example mining mechanism.
>
> We view our contribution as tailored latent diffusion for compact text-to-3D modeling: (1) the choice of SDF representation offers superior efficiency in detailed shape representation and ray tracing compared to the NeRF; (2) the integration of the DeepSDF objective with the diffusion objective via properly introducing Gaussian noise improves the performance; (3) a comprehensive examination of DeepSDF design choices further boost final performance.
>
> All of the above represent a non-trivial endeavor that contributes to the achievement of our ultimate solution for compactness.
>
>
>
>
>
> ---
> > The ablation study seems to miss experiments that demonstrate the role of hard example mining.
>
> In the submission (09/28/2023), we included the related ablation in Table 4 and further fine-grained results in Table 5. They are both in the appendix.

---

> ### Author Response · Authors · 2023-11-21
> **Author Response (part 2)**
>
> > Recent state-of-the-art methods, such as DiffRF (CVPR 2023), 3DShape2Vecset (SIGGRAPH 2023), and Shap-E (arXiv), have also leveraged the idea of performing diffusion in the latent space of NeRF/SDF/occupancy. This paper lacks an in-depth comparison and discussion concerning these methods.
>
> Incorporating your valuable suggestions, we have included a comparative analysis with Shap-E and 3DShape2Vecset in the table below. As 3DShape2Vecset did not originally support text-conditional generation, we mainly compare its model details. (We tried to extend 3DShape2Vecset, `kl_d512_m512_l8` and `kl_d512_m512_l8_d24_edm`, to support text-conditional generation, but failed to do so. It seems finetuning their model costs a lot of GPU memory, and we can barely support batch size 1 backpropagation.)
>
> ------------
> |                | Latent Length | Encoder Parameters | Decoder Parameters | Speeds (s) |
> | -------------- | ------------- | ------------------ | ------------------ | ---------- |
> | 3DShape2Vecset | 4,096         | 106,116,097        | 164,218,368        | 11.16      |
> | Shap-E         | 1,048,576     | 443,771,357        | 315,692,032        | 16.18      |
> | Ours           | 256           | 3,564,502          | 3,904,640          | 1.87       |
> ------------
>
>
> Based on the table, our proposed model features significantly fewer parameters in both the encoder and diffusion model when compared to 3DShape2Vecset and Shap-E.
> Besides, 3DShape2Vecset employs a fixed predefined resolution (its Section 5.3) and limited input resolution, i.e. 2048 points ([ref1](https://github.com/1zb/3DShape2VecSet/blob/bedbd1091664be8e2409d580c4e1630df1a37c89/main_class_cond.py#L39), [ref2](https://github.com/1zb/3DShape2VecSet/blob/bedbd1091664be8e2409d580c4e1630df1a37c89/main_ae.py#L36)), while our proposed approach offers the distinct advantage of unlimited query resolution.
>
> Unfortunately, we were unable to include DiffRF in this comparison as it has not released its source code. We welcome any additional suggestions or feedback you may have in this regard.
>
> ---
> > The claim on page 8 regarding the lightweight framework outperforming larger models like Shap-E and Point-E ("A comparison with Point-E and Shap-E reveals that a lightweight framework can outperform these larger models with training over limited high-quality 3D-text data in the specific geometrical-text- guided 3D shape generation task.") seems unsubstantiated. To strengthen this claim, the paper should consider re-training or fine-tuning Shap-E/Point-E on the same dataset and comparing results.
>
> Following your suggestions, we re-training Shap-E/Point-E on the same dataset. Here’s our anonymized training codes: https://drive.google.com/drive/folders/1m-qJEbHGVqB-rvp3WjeMkUnuxfY2RHxb?usp=sharing. Some details are listed below.
>
> For shap-E: we leveraged the pre-trained encoder, specifically [this transmitter](https://github.com/openai/shap-e/tree/main/shap_e/models/transmitter), and retrain the latent diffusion model from scratch. We opted for the use of a pre-trained encoder because training an encoder from scratch with our data proved to be unfeasible in achieving satisfactory performance.
>
> For point-E, we retrain the base diffusion model (producing 1,024 points), and use the pre-trained upsampling diffusion model (1024 points -> 4096 points) and the pre-trained point-to-sdf model (the regression-based model mentioned in Section 4.5 of point-E paper).
>
> Qualitative results are here: https://drive.google.com/file/d/1wfrBTroGIa84e2d4H8r3EiVKPJN1PCVf/view?usp=sharing. It shows the point-E model failed to predict desirable objects with smooth surface even with pre-trained upsampling and point-to-sdf models. Shap-E model get improvements by train from scratch along with the pre-trained encoder.
>
>
> The table below demonstrates that Shap-E (re-train) achieves improvements over Shap-E (pre-train) while maintaining competitive performance compared to Compact-DiffSDF, despite having 100 times more parameter, `(443771357+315692032)/(3564502+3904640)=101.68013796`.
>
> ---
> | ShapeGlot | FID⬇️ | MMD⬇️ | TMD⬆️ |
> | --------- |  ---- | ---- | ---- |
> | Shap-E (pre-train)    |  92.1 | 11.45 | 2.07 |
> | Shap-E (re-train)    |  78.5 | 6.89 | 3.04 |
> | Ours      |  73.4 | 6.42 | 2.87 |
> ----
>
>
>
> It's important to highlight that the original training datasets for Shap-E and Point-E are substantially larger, ranging from 100 to 1,000 times the size of our dataset. Given this considerable difference in data scale, it's essential to acknowledge that fine-tuning with pre-trained weights may not result in a fair comparison.

---

> > ### Comment · Reviewer_jMCT · 2023-11-22
> >
> > Thank you for your detailed rebuttal. I acknowledge the key points and additional experiments you have provided, which address many of my initial concerns, particularly in terms of comparisons and model performance.
> >
> > However, I concur with Reviewer 6FpV regarding the technical novelty of the work. While the introduction of Gaussian perturbations and hard example mining in the context of DeepSDF is noteworthy, it still appears that the broader methodological approach aligns closely with existing paradigms. A more in-depth discussion or demonstration of how these elements uniquely contribute to the advancement in the field would be beneficial.
> >
> > Additionally, while the results presented in https://drive.google.com/file/d/1-GYKMrnuqFQD3J1XkTJt8j0zkxbyF5KB/view?usp=sharing indeed showcase additional diversity within the scope of the model, the fact that the categories are seen in the training set, as mentioned in the authors' reply to Reviewer jSYM, still raises questions about the model's capability for true out-of-domain generalization.

---

> ### Author Response · Authors · 2023-11-22
> **Author Response**
>
> We greatly thank you for sharing your further thoughts with us. The below are our responses with new results.
>
> ---
> > as mentioned in the authors' reply to Reviewer jSYM, still raises questions about the model's capability for true out-of-domain generalization.
>
> To examine the out-of-domain generalization, we adopted the below two types of Nearest Neighbor (the second way is recommended by Reviewer 6FpV):
>
> - For a given generated object, we compute the Chamfer Distance between the generated object and all the training objects, and locate two Nearest Neighbors.
> - For a given text query, we leverage BERT to compute similarities between the text query and all training texts (codes: https://drive.google.com/file/d/1eRVelGzGfpJknW9Fkd_iZZeFy6FTd_dX/view?usp=sharing). We find two most similar text prompt in the training texts and display their corresponding 3D objects.
>
> The results is shown here: https://drive.google.com/file/d/1mRRQ9O9NQKUI7lVhKRXDykmnhmdtwKWV/view?usp=sharing
> Both the two types of NN can tell our model obtains out-of-domain generalization to some levels.
>
> We welcome your insights on how to enhance the evaluation of out-of-domain generalization, in case you feel that our current approach may not be sufficient to substantiate our claim. **Researchers can consider either our methodology or your suggested approach to ensure a robust evaluation of out-of-domain generalization for Text-to-3D models, which is critical to our community.**
>
> ---
> > while the results presented in https://drive.google.com/file/d/1-GYKMrnuqFQD3J1XkTJt8j0zkxbyF5KB/view?usp=sharing indeed showcase additional diversity within the scope of the model, the fact that the categories are seen in the training set
>
> We further provide more results on Objaverse-Cap3D. Please kindly refer to https://drive.google.com/file/d/1CnzMd5MOngOxYHo_27suHdQCmDKOV_O5/view?usp=sharing. The results of the first column and the first row can show the compositional generalizability of our model to some levels.
>
> ---
> > However, I concur with Reviewer 6FpV regarding the technical novelty of the work. While the introduction of Gaussian perturbations and hard example mining in the context of DeepSDF is noteworthy, it still appears that the broader methodological approach aligns closely with existing paradigms. A more in-depth discussion or demonstration of how these elements uniquely contribute to the advancement in the field would be beneficial.
>
> We won't bother you further on the technical novelty aspects, but we hope you can recognize the contribution and our dedicated efforts in achieving a compact solution for text-to-3D models.

---

### Official Review · Reviewer_6FpV · 2023-10-31

**Soundness:** 3 good
**Presentation:** 3 good
**Contribution:** 3 good
**Rating:** 5
**Confidence:** 4

**Summary:**

This paper focuses on efficient Text-to-Shape generation and proposes a a lightweight model called CDiffSDF. The framework works by learning a diffusion model to predict a latent code from a query text embedding; the latent code is then used to predict SDF samples for the shape, which can be further processed into mesh or point cloud. The model is trained and tested on a text-3D-paired dataset containing over 20,000 shapes. Experiments show that CDiffSDF performs competitively with state-of-the-arts both quantitatively and qualitatively while being compact.

**Strengths:**

1. This paper works on the important task of text-to-3D-geometry generation with a focus on designing a lightweight model. Experiments demonstrate that CDiffSDF performs competitively with state-of-the-arts both quantitatively (Tab. 1.a) and qualitatively (Fig. 3) while being compact (Tab. 1.b).

2. The content of the paper is comprehensive, including extensive experiments and ablation studies, sufficient analysis of limitations, as well as  past failed attempts on voxel-based SDF text-to-shape (supp. B).

3. The paper is well-written and easy to follow and understand. Implementation details are well provided.

**Weaknesses:**

1. Most strategies introduced in Sec. 3.3 lack enough novelty.

(1) Augmenting the latent space with Gaussian noise to improve the 3D shape generation quality of an implicit decoder is not new. Example prior work:
[a] Sanghi et al. CLIP-Forge: Towards Zero-Shot Text-to-Shape Generation. CVPR 2022.
[b] Park et al. Deepsdf: Learning continuous signed distance functions for shape representation. CVPR 2019.

(2) Using the top-k highest losses for hard example mining is not new. Please consider cite related work and discuss the difference. Example prior work:
[a] Fan et al. Learning with Average Top-k Loss. NeurIPS 2016.
[b] Yu et al. Loss Rank Mining: A General Hard Example Mining Method for Real-time Detectors. IJCNN. 2018.

2. The ABO dataset used in the paper contains multiple categories other than chairs and tables. The paper has only shown results of shape reconstruction on multiple categories and text-to-3D generation on chair-like&table. It would be much better to help understand the performance if  the authors could also include some results of text-to-3D generation on other non-chair-like/table categories, eg, lamp, dresser, etc., (although the quality might be a bit worse than chairs/tables due to fewer training examples).

**Questions:**

In supp. fig. 10, the authors have inspected the top-2 nearest neighbors of the generated shapes in training datasets, which is good. Another question is: how does the testing query text differ from the training texts? For example, what are the training text queries that are close to this testing query (eg, inlcluding the testing keywords) and the 3D shapes corresponding to the closest training text queries?

---

> ### Author Response · Authors · 2023-11-22
> **Author Response**
>
> We sincerely appreciate Reviewer 6FpV for the valuable comments and insightful suggestions.
>
> ---
> > The ABO dataset used in the paper contains multiple categories other than chairs and tables. The paper has only shown results of shape reconstruction on multiple categories and text-to-3D generation on chair-like&table. It would be much better to help understand the performance if the authors could also include some results of text-to-3D generation on other non-chair-like/table categories, eg, lamp, dresser, etc., (although the quality might be a bit worse than chairs/tables due to fewer training examples).
>
> Thanks, we’ve generated more results on other categories during the rebuttal! We trained a model with 3D-text pairs from a small subset from Objaverse-Cap3D datasets [1,2]. Please refer to https://drive.google.com/file/d/1-GYKMrnuqFQD3J1XkTJt8j0zkxbyF5KB/view?usp=sharing.
>
> [1]Objaverse: A universe of annotated 3d objects, CVPR23
>
> [2]Scalable 3D Captioning with Pretrained Models, NeurIPS23
>
> ---
> > Most strategies introduced in Sec. 3.3 lack enough novelty.
> (1) Augmenting the latent space with Gaussian noise to improve the 3D shape generation quality of an implicit decoder is not new. Example prior work: [a] Sanghi et al. CLIP-Forge: Towards Zero-Shot Text-to-Shape Generation. CVPR 2022. [b] Park et al. Deepsdf: Learning continuous signed distance functions for shape representation. CVPR 2019.
> (2) Using the top-k highest losses for hard example mining is not new. Please consider cite related work and discuss the difference. Example prior work: [a] Fan et al. Learning with Average Top-k Loss. NeurIPS 2016. [b] Yu et al. Loss Rank Mining: A General Hard Example Mining Method for Real-time Detectors. IJCNN. 2018.
>
>
> Thank you for bringing these relevant works to our attention! We will incorporate a discussion of these works into our revision and provide proper citations.
>
> We've explored the use of Gaussian noise for 3D shape generation, as demonstrated in DeepSDF (equation (1) in our paper). Notably, we've extended this approach to better align with the diffusion objective, as outlined in equation (4) in our paper, which is new. Additionally, it's important to highlight that the effective integration of hard example mining and latent-code-based SDF reconstruction is both new and vital to our ultimate solution for compactness.
>
> ---
> > In supp. fig. 10, the authors have inspected the top-2 nearest neighbors of the generated shapes in training datasets, which is good. Another question is: how does the testing query text differ from the training texts? For example, what are the training text queries that are close to this testing query (eg, inlcluding the testing keywords) and the 3D shapes corresponding to the closest training text queries?
>
> Thanks for raising this suggestion, which complete the top-2 nearest neighbor experiment! We are working on it and will update soon.

---

> ### Comment · Reviewer_6FpV · 2023-11-22
>
> I appreciate the authors' efforts in trying to address the concerns from all reviewers. I have read through all reviewer comments as well as the rebuttals and references from the authors.
>
> 1. Additional experiment on Objaverse-Cap3D:
>
> I was asking for more testing results of the model trained on the mixed dataset as described in the submission, so I don't think this additional experiment solved my question. Furthermore, the new testing results only include very simple (i.e., almost single-world) text queries while the Objaverse-Cap3D dataset contains much richer desciptions -- therefore, I am not convinced by the new results either.
>
> 2. The rebuttal has not solved my concerns about the novelty which have also been expressed by reviewer jMCT.
>
> 3. The authors didn't address my question about the top-2 nearest query neighbor.
>
> Based on all these unsolved issues above, I now have more concerns about the actual performance of the proposed method and the potential contribution of this paper to the community. Therefore, I unfortunately decided to slightly lower my rating.

---

> ### Author Response · Authors · 2023-11-22
> **Author Response**
>
> We sincerely appreciate you for sharing your further concerns, and we can take them into considerations during the rebuttal. We are sorry for the delay in the top-2 nearest query neighbor experiments, as computational resources have been in high demand during the CVPR period.
>
> **As we are currently in the discussion phase, we kindly hope that you postpone making a final decision until you have had the opportunity to review all of our updated results.**
>
> ----
> > I was asking for more testing results of the model trained on the mixed dataset as described in the submission, so I don't think this additional experiment solved my question.
>
> Sorry for the confusion. As for the new results we've presented, the text prompts that do not include the phrase `3D model of` are additional test results from the model submitted earlier (exactly as you ask for). To make this clear, we make a separate illustration https://drive.google.com/file/d/1wM9mHzUehW_FYB8ITVUZaB8Byx8JI16O/view?usp=sharing.
>
> ---
> > Furthermore, the new testing results only include very simple (i.e., almost single-world) text queries while the Objaverse-Cap3D dataset contains much richer desciptions -- therefore, I am not convinced by the new results either.
>
> We generate more interesting results here: https://drive.google.com/file/d/1CnzMd5MOngOxYHo_27suHdQCmDKOV_O5/view?usp=sharing. Please refer to the link and take a look at the first row and column, which indicates some of the compositional
> generalization performance.
>
> As we train on a small subset of Objaverse-Cap3D datasets, our model is expected not covering a lot of complex text prompts. However, if you want to examine certain complex text prompts, we are happy take them into consideration.
>
>
> ---
> > The rebuttal has not solved my concerns about the novelty which have also been expressed by reviewer jMCT.
>
> Since you've confirm our technique contributions, we prefer not to bother you further on the technique novelty aspects. But we hope you can recognize our contribution to the compact Text-to-3D field within our community, including [1] the raise of concern on the compactness of Text-to-3D methods, [2]a comprehensive compact solution, and [3] detailed comparisons with 10+ existing methods.
>
> ---
> > The authors didn't address my question about the top-2 nearest query neighbor.
>
> The results is finally out! Please refer to: https://drive.google.com/file/d/1mRRQ9O9NQKUI7lVhKRXDykmnhmdtwKWV/view?usp=sharing
>
> We leverage BERT to compute the two nearest neighbor of training text queries that are close to this testing query (including the testing keywords). Please refer to our code here: https://drive.google.com/file/d/1eRVelGzGfpJknW9Fkd_iZZeFy6FTd_dX/view?usp=sharing. For the nearest-neighbor text, we list its corresponding object below the text prompt. The results demonstrate that our model indeed generates distinct features compared to the training objects.
>
> We thank you for proposing the second type of Nearest-Neighbor (NN), which use NN text-prompt to identify the generalization performance of text-to-3D model. We believe this kind of evaluation would be generally adopted to evaluate text-to-3D model.

---

### Meta-Review · Area_Chair_VUwF · 2023-12-06

**Metareview:**

The paper proposes a lightweight model for text-to-3D object generation. The paper received two positive and two negative ratings. The positive reviews are based on the well-written presentation and comprehensive experiments. The negative reviews focused on the technical contributions and the out-of-domain generalization. Methodologically, the proposed pipeline is closely aligned to existing efforts, and the proposed components are not significant enough to serve as serve as independent contributions given the scope of experiments. As for the OOD generalization, the results provided in the main paper and the rebuttal cannot fully showcase the robustness of handling OOD use cases. After discussions, we suggest the rejection of the paper and highly encourage the authors to further improve the paper accordingly.

**Justification For Why Not Higher Score:**

Insignificant technical contributions and the out-of-domain generalization ability are the major concerns.

**Justification For Why Not Lower Score:**

N/A

---

### Decision · Program_Chairs · 2024-01-16

Reject